# Engineering the Bridge between Innate and Adaptive Immunity for Cancer Immunotherapy: Focus on γδ T and NK Cells

**DOI:** 10.3390/cells9081757

**Published:** 2020-07-22

**Authors:** Fabio Morandi, Mahboubeh Yazdanifar, Claudia Cocco, Alice Bertaina, Irma Airoldi

**Affiliations:** 1Stem Cell Laboratory and Cell Therapy Center, IRCCS Istituto Giannina Gaslini, Via G. Gaslini, 516147 Genova, Italy; fabiomorandi@gaslini.org (F.M.); claudiacocco@gaslini.org (C.C.); 2Stem Cell Transplantation and Regenerative Medicine, Department of Pediatrics, Stanford University School of Medicine, Palo Alto, CA 94305, USA; myazdani@stanford.edu

**Keywords:** gamma delta T cells, NK cells, γδT cells, immunotherapy, adoptive cell therapy, GvHD, engineered T cell, CAR-T cell, TCR transfer

## Abstract

Most studies on genetic engineering technologies for cancer immunotherapy based on allogeneic donors have focused on adaptive immunity. However, the main limitation of such approaches is that they can lead to severe graft-versus-host disease (GvHD). An alternative approach would bolster innate immunity by relying on the natural tropism of some subsets of the innate immune system, such as γδ T and natural killer (NK) cells, for the tumor microenvironment and their ability to kill in a major histocompatibility complex (MHC)-independent manner. γδ T and NK cells have the unique ability to bridge innate and adaptive immunity while responding to a broad range of tumors. Considering these properties, γδ T and NK cells represent ideal sources for developing allogeneic cell therapies. Recently, significant efforts have been made to exploit the intrinsic anti-tumor capacity of these cells for treating hematologic and solid malignancies using genetic engineering approaches such as chimeric antigen receptor (CAR) and T cell receptor (TCR). Here, we review over 30 studies on these two approaches that use γδ T and NK cells in adoptive cell therapy (ACT) for treating cancer. Based on those studies, we propose several promising strategies to optimize the clinical translation of these approaches.

## 1. Introduction

In recent decades, mainly due to the development of immunotherapeutic approaches that can efficiently target tumor cells, important progress has been achieved in cancer treatment. Initially, cancer’s immunotherapy was based on the use of tumor-infiltrating T lymphocytes (TILs) isolated from tumors, ex vivo expanded and then re-infused into lympho-depleted patients [1]. This form of immunotherapy is called adoptive cell therapy (ACT) and includes the infusion into the patients of ex vivo expanded immune cells. Although cancer regression was observed in patients with solid tumors [2,3,4], such approach posed some difficulties related to (i) challenges of isolating enough number of autologous TILs, (ii) the requirement of reactive T cell population and (iii) a long expansion period [5]. Thus, novel strategies switched the attention to peripheral blood (PB) lymphocytes as source for ACT. PB lymphocytes have been engineered to improve T cell specificity to tumor antigens or tumor-associated antigens (TAA), and such approach revolutionized ACT field. More recently, ACT therapeutic efficacy (e.g., tumor regression and clinical responses) was (and are actually) increased by implementing functionality, expansion, persistence, homing and resistance to inhibitory signals [6]. From this perspective, T cell receptor (TCR) editing and chimeric antigen receptor (CAR) represent two of the most efficacious methodologies that have been clinically tested in cancer therapy. Here we focus on γδ T and NK cells that represent ideal sources for developing allogeneic cell therapies.

γδ T cells and NK cells represent an appealing source for adoptive cell immunotherapy due to their unique biology. The following features pinpoint the favorable characteristics of these cells over αβ T cells for cancer treatment.

First, NK and γδ T cell tumor recognition and killing is not dependent on the expression of a single antigen. In contrast, they recognize a broad spectrum of antigens on various cancer cells through their diverse innate cytotoxicity receptors expressed on their cell membrane [7,8,9,10]. This broad response reduces the chances of tumor immune escape by single antigen loss. In addition, this property provides opportunity for designing immunotherapies for tumors lacking well-defined neo-antigens and without the need for further genetic engineering.

Second, γδ T and NK cells recognize their target cells in a classical MHC-independent manner leading to low or absent risk of alloreactivity and GvHD, thus allowing the development of universal third-party allogeneic cell products for several malignancies.

Third, γδ T cells home in a wide variety of tissues wherein they can rapidly respond to the target and release effector cytokines. This natural tissue tropism of γδ T cells, especially of the Vδ1 subset, provides migratory advantage over αβ T cells and higher ability to infiltrate and function in tumors hypoxic environments [11].

Furthermore, growing evidence indicates that γδ T cells interact with antigen presenting cells (APCs) and other immune cells, while also playing the role of APCs by priming the antigens for αβ T cells thereby enabling the orchestration of a cascade of immune responses against tumors [12]. These features make unmodified NK and γδ T cells an attractive source for ACT. However, genetic engineering strategies may also be applied to enhance their cytotoxicity and redirect them toward specific targets. For example, using γδ T cells, either as a vehicle for CARs or αβ T cell-derived TCRs, [13] may provide exciting results by combining tissue resident property and innate-like recognition of γδ T cells with antigen-specific activation and engagement of multiple co-stimulatory signals. To date, the major obstacle to the broad application of NK and γδ T cells for adoptive cell immunotherapy remains effective strategies of in vivo or ex vivo expansion [14,15].

## 2. Adoptive Cell Therapy Based on TCR Editing and CAR

### 2.1. TCR-Engineered T Cells

ACT underwent a strong methodological evolution starting from TILs, through TCR editing until the most revolutionary platform of CAR-T cells that accounts of a high number of clinical trials worldwide [16].

TCR editing technology is based on the adaptive transfer of transgenic T cells expressing a tumor antigen-specific exogenous TCRαβ TCR gene transfer was first set up by Dembic et al. [17] and was explored by many researchers to redirect αβ T cell response against antigens from viral [18] and tumor targets [19]. Promising results have been obtained in clinical settings in patients with (i) metastatic melanoma infused with PB lymphocytes transduced with a retroviral vector expressing MART-1 TCR α and β chains [20] and (ii) multiple myeloma (MM) treated with CD3^+^ T cells transduced with a lentiviral vector expressing affinity-enhanced NY-ESO TCR [21].

One concern with TCR engineering is that transferred αβ genes can mis-pair with the endogenous αβ chains [22,23,24] which can lead to inefficient expression of the novel construct [25], and generation of hybrid and self-reactive TCR leading to off-target toxicity [22,23,26]. To prevent this, several strategies have been devised including the use of murine constant regions and altered cysteine arrangements in the donor TCRs [26]. The introduction of murinized TCRs runs the risk of mounting an anti-murine immune response in host, which in turn may diminish the clinical efficacy. However, in one study [27], although 23% of patients developed anti-murine-TCR Abs, objective tumor regression, and absence of toxicity were observed.

More recently, novel strategies have been implemented to overcome these pitfalls, such as introducing molecular modifications to TCR α and β chains which improves the correct formation of TCR complexes [26,28,29]. The best results have been achieved by deleting endogenous TCR before the introduction of exogenous tumor-specific TCR chains. For example, Albers et al. replaced the endogenous TCR with the exogenous one in the *T-cell receptor α chain (TRAC) locus*, using CRISPR-Caspase 9 technology. They edited a TCR specific for myeloperoxidase-derived peptides which is a myeloid neoplasia TAA presented by human leukocyte antigen (HLA)-B7 molecule, and reported that these lymphocytes selectively targeted tumor cells in the pre-clinical models [30,31]. Similar results were obtained by Roth et al. who adopted the CRISPR-Caspase 9 strategy to replace the endogenous TCR with a NY-ESO specific exogenous TCR. In this case, transduction of T cells was performed by electroporation instead of viral vectors. T cells bearing NY-ESO specific TCR were able to recognize the tumor cells, infiltrate the tumor and reduce the tumor growth [32]. Electroporation was also used to transfer γδ TCR or Mel13 TCR (specific for a Melan-A epitope presented in the context of HLA-A2) into T cells. A study [33] showed that ablation of endogenous TCR increased the expression of exogenous TCR and, accordingly, higher cytotoxicity and cytokine production was observed in response to the tumor cells.

A clinical application of the mentioned technique was reported by Stadtmauer [34] who depleted PD-1 and the endogenous TCR in T cells concomitantly transduced with viral vectors encoding NY-ESO specific TCR. The clinical trial showed that (i) engraftment and persistence of edited T cells in patients was higher than those observed in patients with endogenous TCR and PD-1, (ii) edited T cells targeted tumor cells and (iii) injected cells migrated to the tumor site. Provasi et al. developed a novel technique based on the use of zinc-finger nucleases (ZFNs) which can target and eliminate specific DNA sequences. This technique was used to delete both *TRAC* and *T-cell receptor β chain (TRBC) loci* in T cells before transduction with cDNA encoding HLA-A2-restricted TCR αβ chains specific to Wilm’s tumor antigen (WT) 1 peptide [35]. Such approach was useful not only to deplete endogenous TCR αβ chains, but also to enhance avidity and specific killing against tumor cells in vitro. More importantly, pre-clinical studies demonstrated that alloreactivity of T cells was almost abrogated when endogenous TCR was depleted, as witnessed by the absence of graft-versus-host disease (GvHD) [36].

### 2.2. CAR Engineered T Cells

Initial studies addressing the potential use of chimeric receptors were published by Kuwana et al. [37] and Gross et al. [38] wherein the immunoglobulin-derived V regions and T cell receptor-derived C regions were implemented. Since then, the possibility to generate and express chimeric receptors into T cells incorporating a signaling moiety has been studied and resulted in successful implementation of CAR [39].

CAR constructs are composed of (i) the single-chain variable fragment (scFv) of tumor antigen-specific Ab (Ab), (ii) a hinge region, (iii) the hydrophobic trans-membrane domain which is usually derived from CD8α or CD28 and (iv) the intracellular signaling moieties. All these parts are necessary to optimize and maximize T cell antigen recognition, T cell activation, and tumor cell lysis. Based on the number of intracellular domains, CARs are classified into 1st, 2nd, and 3rd generations containing one, two, or more T cell co-stimulatory molecules, respectively [40,41]. Originally, the 1st generation CAR only consisted of the activating/signaling moieties of CD3ζ or FcγRIII endodomains. The 2nd generation CAR had a co-stimulatory domain belonging to molecules of CD28 family, such as CD28 and Inducible T cell costimulator (ICOS), or TNF receptor family (4-1BB, OX-40 and CD27), whereas the 3rd generation made use of multiple co-stimulatory domains in tandem (i.e., CD28 in combination with 4-1BB), which in some cases increased the expansion and anti-tumor activities [42].

The outstanding success of CAR-T cell clinical trials led to Food and Drug Administration (FDA) approval of CAR-T therapy in 2017, in particular of two products, namely Kymriah^®^ (Novartis) and Yescarta^®^ (Kite/Gilead), for the treatment of hematological malignancies [43]. These clinical trials gave exciting results which were significantly different from those observed in more than 100 clinical studies on solid tumors [44].

### 2.3. CAR Versus TCR: Pros and Cons

As earlier reported, many studies highlight the potential of TCR editing as a therapeutic strategy for patients with hematologic and solid cancers with limited side effects. CAR technology has also been undoubtedly successful especially in treating B cells neoplasms; however, it has raised major toxicity concerns. Both approaches have pros and cons which we discuss here.

First, TCR edited T cells can recognize a variety of intracellular and surface antigens degraded by proteasome and presented by major histocompatibility molecule (MHC). This characteristic expands the range of detectable antigens; however, it limits the application of TCR edited T cells due to the requirement of MHC-restricted recognition and co-stimulation [45]. In contrast, CAR-T cells recognize only surface native antigens in an MHC-independent manner, and does not need additional co-stimulation which is a clear advantage of CAR application [46]. Conversely, TCR editing approach, compared to CAR-T, is associated with lower rates of cytokine release syndrome (CRS) and neurotoxicity, the two of the more frequent and severe complications observed in CAR-T cell therapies [47]. However, infusion of CAR-T cells provides better clinical outcomes in patients with hematological malignancies. This is conceivably related to different mechanisms of activation and/or antigen recognition. In fact, CAR activation and functions are independent from TCR and not subjected to regulatory mechanisms. By contrast, TCR edited T cells use the same activating and regulatory circuits of natural TCR [48].

Some studies proposed the possibility to design a system combining the flexibility of CARs and the ability of TCR edited T cells to recognize a wide variety of antigens, using TCR-like Ab fragments or CAR/TCR hybrids. These molecules can be isolated from phage display libraries and further converted into CARs [49,50].

The outstanding success of engineered T cell therapy for the hematological malignancies has encouraged scientists to use this technology for the treatment of solid tumors and to develop allogenic “off-the-shelf” therapies. TCR γδ T cells and NK cells hold unique characteristics which appoints them as suitable sources for fulfilling these goals. γδ T and NK cells are two cell subsets that belong to the innate immunity and recognize different molecules expressed on neoplastic cells in a manner that is independent of classical MHC. Both γδ T and NK cells express NKG2D molecule which interacts with non-classical MHC molecules such as MHC-I chain-related (MIC) peptides A and B (MICA/B) expressed on stressed cells. This mechanism contributes to tumor immune surveillance [51]. These characteristics make unmodified γδ T cells and NK cells an attractive source for adoptive immunotherapy. However, genetic engineering strategies may be applied to enhance their cytotoxicity and redirect them toward specific targets. Ultimately, γδ T cells and NK cells can be used to develop universal off-the-shelf cell therapy for cancer overcoming the challenges of allogeneic therapies. In addition, either specific or broad anti-tumor activities may be simultaneously achieved using these cells as ACT. In this review, we focus on discussing the application of genetic engineering strategies such as TCR and CAR in γδ T and NK cells, as well as the pre-clinical and clinical so far data obtained from their translation into patients.

## 3. γδ T Cell Engineering Strategies

### 3.1. TCR Gene Transfer Involving γδ T Cells

As previously mentioned, an alternative method to avoid TCR chains mispairing and GvHD occurrence is represented using γδ T cells. Since γ and δ TCR chains do not pair with the transferred α and β chains, an entirely human αβ TCRs can be transferred into γδ T cells [52]. Similarly, γδ T cells may be also used as recipients of TCR expressed by invariant natural killer T (iNKT) cells. Conversely, γδ TCR gene transfer into αβ T cells endows αβ T cells with γδ T cell-like properties. The latter approach combines the knowledge of targeting cancer through individual receptors expressed on γδ T cells with the high proliferation and memory capacity of conventional αβ T cells, overcoming the issue that γδ T cells are frequently deleted in patients with advanced-stage disease, limited in their proliferation ability and prone to produce IL-17, thus contributing to tumor tolerance [53].

αβ T cells transduced with γδ TCRs can target a broad range of solid and hematological tumors, gaining advantage from the fact that activating signaling through the γδ TCR is less prone to inhibition by Killer Inhibitory Receptors (KIRs), owing to the lower abundance of such receptors on αβ T cells than on γδ T cells. Furthermore, CD4^+^ and CD8^+^T cells transferred with γδ TCR maintained cytotoxic properties and have the ability to mature dendritic cells (DCs) that in turn, sense increased phosphoantigen levels on immature DCs in the presence of aminobisphosphonates through the γδ TCR [53]. Finally, replacement of αβ with γδ TCR overcomes GvHD development in patients.

Conversely, γδ T cells transduced with αβ TCR acquire a superior proliferation potential, tumor antigen specificity, while maintaining their intrinsic homing ability and avoiding GvHD.

Three main strategies of TCR gene transfer using γδ T cells as carrier or as TCR donor are reviewed below.

### 3.2. αβ TCR Transfer into γδ T Cells

Studies showed that γδ T cells transduced with a HLA-A0101 restricted αβ TCR produced more interferon (IFN)-γ and TNF-α than CD8 αβ T cells expressing the same TCR, but had comparable cytotoxicity against target cells [54]. Similar results were attained when γδ T cells were RNA-transduced to express a gp100/HLA-A2 restricted αβ TCR and subsequently exposed to gp100^+^ melanoma cells. These γδ T cells were also found to preserve their intrinsic anti-tumor function and ability to lyse MHC-deficient cells [55].

Co-transferring the αβ TCR and co-activator receptors, such as CD4 and CD8, has been examined by few researchers. A study showed that potent anti-leukemic activity and cytokine production of αβ TCR-transduced γδ T cells were further enhanced once CD4 or CD8 molecules were co-transferred [56]. Similarly, Hiasa et al. demonstrated that γδ T cells co-transduced with αβ TCR and CD8 gene gained robust anti-tumor activities and release of cytokines in both αβ- and γδ TCR-dependent manners. These αβ TCR and CD8-transduced γδ T cells reacted more rapidly to the target cells compared to the conventional αβ T cells [57].

One major limitation related to the transduction of αβ TCR is the restriction to a particular HLA type. This hurdle can be overcome by using the HLA-unrestricted TCRs, such as γδ TCR or TCRs derived from iNKTcells [58].

### 3.3. γδ TCR Transfer into αβ T Cells

Marcu-Malina et al. demonstrated that αβ T cells expressing the Vγ9Vδ2 TCR clone G115 [58] displayed similar native functional properties of the Vγ9Vδ2 cells [59]. Those properties included cytotoxicity against Daudi cell line, secretion of TNFα and IFN-γ, improved cytotoxicity following pre-treatment with phospho-antigen (pAg), and the ability to induce DC maturation. Interestingly, Vγ9Vδ2 transduced into αβ T cells did not launch an alloreactive response linked to down-regulation of their endogenous αβ TCRs [59], and were cytotoxic against a broad panel of cancer cell lines. These findings are in agreement with the physiological role of both Vδ1 and Vγ9Vδ2 γδ T cell subsets that are avoided of alloreactivity against normal cells [60].

Studies have shown that Vγ9Vδ2 TCR derived from different T cell clones, elicits different anti-tumor responses which were not correlated with the expression of NKG2D, CD158a, NKAT-2, or NKB-1 [60,61]. This property offers the opportunity to modify the functional avidity of interaction between Vγ9Vδ2 TCR and BTN3A1 as a rich area for optimization. Transferring Vγ9Vδ2 TCR into CD4 αβ T cells as the recipient, revealed that the length and sequences of CDR3 motifs are critical in ligand interaction and shaping the physiologic Vγ9Vδ2 T cell repertoire [61]. Highly optimized γδ TCRs can be expressed in more readily available αβ T cells to develop adaptive T cell therapy. Straetemans et al. [62] developed an enrichment method for the production of untouched engineered immune cells, easy to be translated into a GMP-grade process and potentially applicable to all receptor-modified cells. It was based on the rationale that following retroviral transduction, non-engineered and poorly engineered immune cells characterized by a high endogenous αβ TCR expression, may be efficiently depleted with GMP-grade anti-αβ TCR beads. The untouched enrichment of engineered γδ T lymphocytes translated into highly purified receptor-engineered cells with strong anti-tumor reactivity both in vitro and in vivo. Such γδ TCR-engineered αβ T cells, prevented tumor growth in an immune-deficient mouse model of MM and Burkitt lymphoma and protected mice from re-challenge with myeloma cells [62]. Importantly, this approach eliminated residual alloreactivity of engineered immune cells.

### 3.4. iNKT TCR Transfer Into γδ T Cells

Similar to γδ T cells, iNKT cells are a rare subset of T cells bridging innate and adaptive immunity. iNKT cells share properties of both T cells and NK cells and express a highly restricted T cell receptor which recognizes glycolipid antigens presented by CD1d [63]. They are activated early in an immune response and therefore can influence differentiation, polarization, and activation of other immune cells including DC, B, NK, and T cells. After activation, iNKT cells rapidly produce large quantities of Th1 and Th2 cytokines such as IFNγ, IL-4, GM-CSF, IL-2, and TNF-β [64]. Simultaneously, they can exert cytotoxic effects against tumor cells via perforin/granzyme and FasL [65].

Beside their immunomodulatory roles, iNKT cells have some important effector functions and play a key role in immune response including host defense, malignancies and inflammation [66,67]. In the context of cancer, it has been reported that their frequency in the tumor or in the circulation may be highly correlated with overall survival in numerous human cancers [68,69,70]. This conceivably related to anti-tumor effects mainly mediated by cytotoxic functions of infiltrating iNKT cells. However, their optimal activation requires DC loaded with iNKT cell ligand such as α-galactoeramide (α-GalCer), i.e., the first glycosphingolipid ligand discovered for iNKT cells [71]. Based on this knowledge, most clinical trials have been conducted using administration of α-GalCer-pulsed CD1d^+^ DCs (DC/Gal) in solid tumors and hematological malignancies that were well tolerated in patients [72].

Such therapeutic approach increased the frequency of iNKT cells capable of producing high levels of IFN-γ, reduced metastases and showed positive clinical responses especially in MM patients [73,74,75].

The use of iNKT TCR in cancer immunotherapy has shown promising results. Transfer of iNKT-derived TCR into γδ T cells led to γδ T cells expansion in the presence of glycolipids and increase of cytotoxicity against the CD1d expressing leukemia cell line. iNKT TCR-transduced γδ T cells retained their γδ TCR expression and were able to respond to γδ T cell ligands. This approach which produces a bi-potential innate lymphocyte can be exploited for treating patients with various tumors [76].

### 3.5. Other TCR Transfers

In a recent study, a comprehensive RNA-seq analysis was used to identify the sequence of γ and δ chains in triple negative breast cancer (TNBC) infiltrating γδ T cells. By reconstructing the TCR γδ genes derived from TILs and engineering αβ T, γδ T and NKT cells, a potent anti-tumor reactivity against TNBC and different cancer cell types was achieved. The response against tumors was dependent on both TCR γ and TCR δ chains and independent of additional co-stimulation via other innate immune receptors. Hence, they underlined the intrinsic ability of γδ T cells in recognizing cancer cells through their individual TCR γ and δ chains. Identification of such tumor sensing TCRs provides new opportunity for designing universal anti-cancer T cell therapy [77].

All the aforementioned evidence delineates the importance of γδ T cells in TCR gene transfer strategies both as TCR donor and recipient cells. Lack of TCR chains mispairing when combined with αβ TCR, non-HLA-restriction and lack of alloreactivity qualify γδ T cells as a promising source for TCR engineering cancer therapies (Figure 1).

### 3.6. CAR Engineered γδ T Cells

#### 3.6.1. Why Engineer γδ T Cells with CAR?

Conventional approaches using CAR αβ T cells resulted in impressive clinical responses in leukemia but have yet to be translated to equivalent successes in solid cancers. Successful treatment of solid tumors depends on the ability of CAR-T cells to home to the tumor site, overcome the immunosuppressive tumor microenvironment, and persist long term. Moreover, CAR αβ T cells show unwanted side effects including CRS and GvHD.

Beyond intrinsic features already mentioned, CAR γδ T cells represent a novel tool due to their additional desirable functional properties. Specifically, CAR γδ T cells, as a first line of defense, are already primed for innate cytotoxicity, recognize a wide range of antigens [57], have natural residence in the solid tumor microenvironment and may acquire phenotype and properties of professional APCs [78,79].

Although γδ T cells are difficult to expand efficiently ex vivo, it is feasible to obtain sufficient number for adoptive transfer immunotherapy for cancer and have potent tumor antigen-dependent cytotoxicity. Their capacity for migration and for uptake and cross-presentation of TAAs marks them as having potential advantages over conventional CAR-T cells, especially in the solid tumor setting. Results from early clinical trials have proven safety and shown promising results, although potency is disparate probably due to the lack of optimal expansion and long-term survival of γδ T cells in patients [80,81,82,83,84]. Therefore, using modern immunotherapy techniques to increase survival and potency by manipulating cellular behaviors remains highly attractive. γδ T cells naturally migrate to the tumor site and CAR transduction might enhance their cytotoxicity without affecting their migratory capability which is beneficial for treating solid tumors. In addition, γδ T cells cross-presenting the tumor antigens at the tumor site, can prolong the intra-tumoral immune response [13].

γδ T cells also express a series of co-stimulatory molecules including CD28, CD27 and 4-1BB which increase their activation and effector functions [85,86,87]. Thus, CAR expression in γδ T cells may benefit from the synergistic effect with other co-stimulatory molecules.

Genetic engineering of γδ T cells is of particular relevance when γδ T cells response is minimal against a specific tumor. Neuroblastoma is a typical example [60] since the tumor shedding soluble NKG2D ligands blocks NKG2D activation [88]. In this case, the use of genetic approaches to engineer γδ T cells can reinvigorate and redirect their cell function. With this in mind, it might be useful to provide additional stimuli that can restore γδ T cell cytotoxicity via a conventional CAR [13,89], opsonization of target cell [60,90], or by restoring the NKG2D signal using a chimeric co-stimulatory receptor (CCR) [89].

It is known that γδ T cells can uptake, degrade, and load tumor antigens on the MHC-I molecules but the underlying mechanisms are not well defined. Studies have shown that this cross-presentation involves insulin-regulated aminopeptidase (IRAP)–positive early and late endosomes and cytosolic proteasome-dependent cross-presentation pathway [91]. A series of studies have reported that γδ T cells expressing CAR retain the ability to cross-present the tumor antigens [13,92,93,94]. Capsomidis et al. demonstrated that γδ T cells transduced with a 2nd generation anti-GD2 CAR (GD2-28ζ) successfully migrate toward the tumors, take up tumor antigens and cross-present the processed peptides to responder αβ T cells leading to their clonal expansion. A 25 amino acid long peptide of the melanoma antigen MART-1, containing epitopes too long to be MHC-presented, was used to pulse the HLA-A201^+^V2δ^+^GD2-28ζ^+^ cells. In this way, the cells resulted able to stimulate secondary expansions of αβ T cells expressing an HLA-A201-restricted MART-1. This ability could be exploited for priming a diverse population of autologous αβ T cells against many tumor antigens [13].

#### 3.6.2. CAR γδ T Cells Engineering Strategies

There are very few studies on CAR γδ T cells compared to the plethora of literature on CAR αβ T cells. The first study to engineer γδ T cells with CAR was published in 2004 [95], when a 1st generation CAR was used to target GD2 (14.G2aζ) antigen on Neuroblastoma and Ewing sarcoma cells. The authors demonstrated that G2aζ^+^Vγ9 T cells, co-cultured with the GD2^+^ Neuroblastoma cell line LAN-1, showed increased antigen-specific tumor reactivity and superior Th1 cytokine production compared to non-transduced zoledronic acid (ZOL)-expanded Vγ9T cells. Accordingly, the T cell activation marker CD69 and INF-α expression was up-regulated in the presence of the tumor cells. This target dependent cytokine production was mirrored in CD19ζ CAR expressing γδ T cells when co-cultured with CD19^+^ cell lines (Daudi, Raji, and Reh) [95,96].

CAR^+^γδ TCR^+^ cells expressing CCR7 and CD62L suggested the ability to home to the bone marrow and lymph nodes where CD19^+^ leukemia reside. When co-cultured with a human CD19^+^ murine cell line, these CAR^+^γδ TCR^+^ cells produced IFN-γ, TNFα, MIP-1α, MIP1β, and CCL5 following CAR activation, but did not engage the γδ TCR due to inter-species differences [97,98]. Moreover, the engineered cells lysed human CD19^+^ cell lines with higher efficiency than CAR^−^γδ TCR^+^ cells. CAR γδ T cell treatment increased the survival of immune-deficient mice bearing CD19^+^ffLuc^+^ NALM6 B cell leukemia, as compared to untreated animals [99].

A recent study has introduced a new strategy to engineer Vδ2 γδ T cells which limited tonic signaling associated with CAR CD3ζ endodomain. A CCR lacking CD3ζ, but containing DAP10 stimulatory domains, was expressed in γδ T cells. Using “AND gate” system, which divides the signals, CCR γδ T cells were activated and released cytokines only when both CD3 and CCR receptors were engaged, whereas in CAR γδ T cells only CAR activation was sufficient. Using single-cell signaling analysis and mass cytometry (CyTOFF), which allowed a detailed characterization of the CAR signaling, they demonstrated that CCR in γδ T cells induced minimal baseline activity of CD3ζ-associated kinases in the absence of the antigen, meaning that tonic signaling decreased and the network plasticity increased [100]. This approach can ameliorate engineered T cells exhaustion and off-target cytotoxicity, features that affect consistently the clinical efficacy of CAR-T cells [40] (Figure 1).

CAR γδ T cells have shown pre-clinical promising data, but the question concerning the optimization of the right co-stimulatory domains remains open [101]. A deeper knowledge of the γδ TCR biology, co-stimulation, and interaction with CAR molecule can provide insight for optimization of CAR γδ T cells. It has been shown that γδ T cells express a series of co-stimulatory molecules such as CD28, CD27, and 4-1BB which once over-expressed, enhanced γδ T cells survival and proliferation. The synergy between γδ TCR and these signals may be explored for clinical expansion of γδ T cells as well as for the design of more suitable CAR constructs [102]. Although engineered γδ T cells have made significant strides in the early phase trials of hematologic malignancies, their clinical efficacy in solid tumors is still uncertain and may be compromised due to the immune suppressive conditions and complexity and heterogeneity of tumor microenvironment [101]. Engineered γδ T cells with receptors rendering them resistant to immune-suppression and combinatorial treatments with agents targeting the stroma and checkpoint inhibitors may improve the treatment efficacy in solid tumors.

#### 3.6.3. Artificial APC/Engineered Feeder Cells

γδ T cells can be expanded ex vivo using pAgs and bisphosphonates; however, these strategies preferentially expand Vδ2 subset [103]. To expand the full repertoire of γδ T cells without a skewed subsets distribution, use of artificial (a)APC or K562 feeder cells is favorable. Since NK cells grow on APCs, their depletion before co-culture is a mandatory step. These aAPCs may be stimulated with pAgs, loaded with anti-γδ TCR Abs [60] or genetically engineered to express a specific antigen in order to enhance the activation and cytotoxicity of γδ T cells or shape their phenotype for specific purposes. aAPCs can be engineered to express a variety of co-stimulatory molecules and cytokines to enable precise determination of the T cells for the downstream therapeutic goals [104]. Deniger et al. were the first group to implement aAPC strategy for expanding polyclonal γδ T cells. They used K562 feeder cells modified to express CD19, CD64, CD86, 4-1BBL, and membrane-bound IL-15, which enabled an expansion of 4900 ± 1700 folds in γδ T cells. In comparison with the previous results obtained from cytokine-based methods [105], this was a remarkable improvement. A different modification of K562 cell line was used by Cho et al. FACS sorted Vγ9Vδ2 T cells were cultured upon CD32, CD83, and 4-1BBL expressing feeder cells. This approach was only able to induce a 12-fold expansion by day 14 and a 106-fold by day 100 which was not comparable with Deniger et al. work [106].

In another study on Vδ2 T cells post-allogeneic hematopoietic stem cell transplant (HSCT), a robust expansion of Vδ2T cells was attained by co-culturing with pAg-pretreated DCs, especially by allogeneic third-party DC. This study confirmed that the frequency and function of DCs are critical for the recovery of Vδ2T cells post-HSCT. In addition, it allows for development of individualized immunotherapy strategies that harness the anti-leukemic and anti-viral effects of γδ T cells in the context of HSCT [107].

Fisher et al. established an aAPC-based approach for unbiased expansion of γδ T cells from peripheral blood mononuclear cells (PBMNCs) of cancer patients. Using aAPC loaded with anti γδ TCR Ab resulted in expansion of three main populations of Vδ2^+^, Vδ1^+^ and Vδ1^−^Vδ2^−^ γδ T cells. Having a more favorable innate killing and memory phenotype, this broad repertoire of γδ TCR may allow both innate killing and Ab dependent cellular cytotoxicity (ADCC) activity to be exploited [60].

In a recent study, Polito et al. [108] used irradiated, engineered aAPCs expressing CD86, 4-1BBL, CD40L, and the CMV-antigen-pp65. CMV-pp65 and CD40L genes are proved to be essential for the expansion of memory Vδ1 subpopulation. To ensure safety and enhance the clinical translation, aAPCs were stably transduced with an inducible suicide gene. The expanded γδ T cells retained polyclonality and potent anti-tumor activity both in vitro and in vivo without sign of alloreactivity, and up-regulated the expression of activation and memory markers, without exhaustion. These cells were further engineered with a 3rd generation anti-GD2 CAR (GD2.CD28.4-1BBζ) using retroviral vectors which enhanced cytotoxicity against neuroblastoma cell line. In sum, their results introduced a complex yet efficient method for producing an allogeneic third-party and ready-to-use, γδ T cell bank (Figure 1).

For advancing to the clinic, feeder cells-based expansion of γδ T cells are not ideal, considering the potential presence of residual feeder cells in the final product. One solution is γ-irradiation of the feeder cells in order to prevent their in vitro growth and unintended proliferation post patient infusion [109,110]. So far, only the presence of an indiscernible levels of γ-irradiated feeder cells after seven days of co-culture have been reported [109]. Since the number of residual feeder cells in the final product is negligible and within the limit assigned by FDA (<0.1%), K562 feeder cell-based expanded immune cells have been approved for clinical trials (NCT00640796 and NCT00995137).

### 3.7. Alternative γδ T Cells Engineering Strategies

An interesting and less common engineering approach was designed to harness γδ T cells endogenous anti-tumor activity to treat resistant glioblastoma (GB). Temozolomide (TMZ), a main chemotherapeutic drug used to treat GB, increases the expression of stress-associated NKG2D ligands on TMZ-resistant glioma cells, potentially rendering them vulnerable to γδ T cell recognition and killing. Since TMZ is highly toxic to γδ T cells, to allow for γδ T cell therapy, γδ T cells were genetically modified using a lentiviral vector encoding the DNA repair enzyme O(6)-alkylguanine DNA alkyltransferase (AGT) which confers resistance to TMZ. This genetic modification resulted in greater cytotoxicity of γδ T cells against TMZ-resistant GB cell lines compared to unmodified γδ T cells, suggesting that TMZ uncovered more receptors for γδ T cell targeted lysis. This study provides the necessary foundation to pursue such innovative drug resistant immunotherapy approaches for treating resistant cancers. Retaining the viability of γδ T cells at high concentrations of TMZ provide an opportunity for synergistic therapy. Delivering high dose cellular immunotherapy directly to the site of residual malignancy when tumors are most vulnerable during chemotherapy can be highly effective [111]. Currently companies such as Incysus Therapeutics are attempting to produce drug resistant γδ T cells for developing allogenic anti-cancer immunotherapy and recently started their first clinical trial on patients with newly diagnosed GB (NCT04165941).

Genetic engineering of T cells with cytokine receptors to enhance their expansion, survival, and function has been attempted by several groups. Orthogonal design (mutated) limits the pleiotropic effect of the cytokines such as IL-2 and reduces the cytotoxicity and off-target effects [112]. Sockolosky et al. demonstrated that engineering CAR αβ T cells with innovatively designed orthogonal IL-2 cytokine receptor pair enhanced the function of CAR αβ T cells in vitro and in vivo [112]. Other cytokines such as IL-7 and IL-15 which are known to have key roles in γδ T cells maintenance and survival [113] may be investigated as appropriate candidates for orthogonal approaches. Different genetic engineering approaches used for using the anti-cancer properties of γδ T cells are depicted in Figure 1.

## 4. NK Cell Engineering Strategies

NK cells are bone marrow-derived innate lymphocytes capable of eliminating the tumors directly. Their activity is governed by the integration of multiple signals from activating and inhibitory receptors as well as from cytokines including IL-15, IL-12, and IL-18 [7,8,9]. Similar to γδ T cells, NK cells do not cause GvHD, thus representing an ideal source for allogeneic “off-the-shelf” cellular therapy. Due to lack of in vivo clonal expansion and limited persistence, CAR-NK cells are less likely to induce CRS, a life-threatening toxicity which has occurred in several CAR αβ T cell trials [114]. On the other hands, persistence of CAR-T and CAR-NK cells may differ according to the nature of the disease, the tumor burden, lymphodepletion settings, and other factors including the potency of the administered T cells and the construct of the CAR, thus representing a drawback of this therapeutic approach. To overcome this issue, some strategies have been implemented such as co-transfection of stimulatory cytokines (e.g., IL-15) and administration of multiple CAR doses at different time points.

Primary NK cells are difficult to isolate, purify, and transduce, and often results in a heterogeneous mixture of cells that poorly expand [115]. As an alternative, the NK cell line, NK92, has been used in clinic, as it can expand easily and indefinitely, but it requires irradiation prior to infusion into patients due to its chromosomal abnormalities and the risk of malignant transformation [116]. In the future, the technology of induced pluripotent stem cells (iPSCs) may offer another renewable source of NK cells facilitating the development of an off-the-shelf therapy.

### 4.1. TCR-Engineered NK Cells

As described for γδ T cells, also NK cells have been investigated for TCR engineering approach. Mensali et al. [117] evaluated for the first time the possibility of transferring an αβ TCR into NK cells, more specifically into NK92 cell line that is FDA-approved, represents a model of universal cells to be useful in allogenic settings, and does not require long-term and expensive culture procedures.

They demonstrated that compared to unmodified NK92 cells, TCR-NK92 cell line showed a novel gene expression pattern that resembled those of primary effector T and NK cells. In fact, TCR-NK92 cells acquired phenotype, metabolism, and functionality of T cells, while maintained typical effector functions of NK cells.

Similarly, another study [118] demonstrated that ectopic expression of αβ TCR in NK92 and YTS NK cells along with CD3 subunits induces the expression of a functional antigen-specific TCR against the HLA-A2 restricted tyrosinasi-derived melanoma epitope Tyr_368–377_. Most importantly, these cells showed anti-tumor killing capabilities against melanoma cells both in vitro and in vivo.

A more particular technique was elaborated by Walseng et al. that consists of endowing NK92 cells with a soluble TCR previously validated by the same group linked to the trans-membrane and signaling domains of a CAR construct, i.e., trans-membrane CD28 associated with intracellular domains of CD28 and CD3ζ. TCR-CAR engineered NK92 cells expanded their specific antigen recognition pattern, enhanced killing activities against target tumor cells and increased production of pro-inflammatory cytokines [50].

### 4.2. Source of NK Cells for CAR Development

NK cells for adaptive cell therapy can be obtained from autologous or allogeneic donors. Their available sources include PB, bone marrow (BM), human embryonic stem cells (hESCs), iPSCs [105,119,120], umbilical cord blood (UCB) and readily available NK cell line [121]. PB- and UCB-derived NK cells were used by Herrera et al. who developed CD19-CAR-tranduced NK cells for the therapy of CD19^+^ leukemias [122]. NK cells were expanded with IL-2 and IL-15, transduced and then cultured before functional assays. The authors showed a higher number of NK cells obtained from UCB compared to PB. However, fold expansion, viability (76–78%) and transduction efficiency (46–47%) were similar in both NK cell populations [122].

Despite these encouraging results, there is several limitations surrounding the use of PB as NK source including risks associated with the apheresis collection of cells from healthy donors’ PB or BM [123,124]. In addition, obtaining NK cells from hESCs or iPSCs requires a complex process [105,119] which ascribes NK cell lines as favorable source. NK cell lines including NK-92 [125,126,127,128,129,130], KHYG-1 [131], NKL [132], NKG, and YT provide an easily accessible and homogeneous source for generation of a large number of CAR-transduced NK cells. NK-92 is a highly cytotoxic NK cell line, originally derived from a patient with NK lymphoma [133]. NK-92 cells express phenotypic and functional characteristic of activated NK cells [133,134]. Phase-I clinical studies demonstrated the safety of NK-92 cell infusion in cancer patients, even at doses of 10^10^ cells/m^2^ [135,136,137]. Due to reported evidence confirming safety, the NK-92 cell line has been broadly used to generate several types of CAR-NK cells, including (i) CD38 nanobodies CAR-NK cells designed to target CD38^+^MM and other hematological malignancies [138], (ii) GD2 CAR-NK cells targeting neuroectodermal tumors [139], (iii) CD19 and CD20 CAR-NK cells which recognize B cell malignancies [128,140], (iv) epithelial cell adhesion molecule (EpCAM) CAR NK cells targeting colorectal cancer [141], (v) GPC3 CAR-NK cells to target hepatocellular carcinoma [142], (vi) CS1 CAR-NK cells to kill MM [143], (vii) epithelial growth-factor receptor (EGFR) CAR NK cells to target breast cancer (BC) [127] and GB [132], and (viii) CD4 CAR-NK cells to counteract T cell leukemias [144]. Tissue factor (TF)-specific CAR-NK cells were generated using a variant of NK-92 cells, named NK-92MI, which in contrast with parental cell line, is not dependent on IL-2 for cell growth. Since such cells lack CD16 expression, Hu et al. performed a first transduction with CD16 followed by a second transduction with TF-specific CAR. Therefore, they obtained CD16^+^TF-CAR-NK cells, capable of mediating IgG1 and IgG3-mediated ADCC. The limitation of such strategy was the low transduction efficiency (~10%) [145]. The two-step transduction approach provided superior transduction efficiency (>90%) when used to generate ErbB2 and CD33 CAR-NK cells [129]. Notably, the CD33 CAR-NK cells generated from NK-92MI cells have been produced in a Good Manufacturing Practice (GMP) facility [146].

### 4.3. NK Cell Transduction with CAR-Encoding Viral Vectors

Lentiviral vectors have been broadly used to generate NK cells expressing CARs with 1st, 2nd and 3rd generation designs which target different tumor antigens including CD19 [122,140], CD20 [140], TF in TNBC [145], CD33 [146], CD7 [147], CD22 [148], ErbB2 [129], EpCAM [141], Glypian-3 (GPC3) [142], CS1 [143], EGFR [127,132], B cell maturation antigen (BCMA) [149] and CD4 [144]. The specific construct contains the scFv region of a tumor antigen-specific Ab, which may be combined with (i) enhancers such as Kozak sequence [145], (ii) signaling domains such as 4-1BB [122,141,144,146] and/or CD28 [127,129,132,142,143,144,146], (iii) CD8α hinge region [141,142], (iv) myc tag sequence [143], and (v) CD3ζ chain of TCR complex [122,127,129,132,140,141,142,143,144]. These constructs are transiently transduced in 293T cells, along with packaging constructs, to produce supernatants enriched with viral particles to be used for NK cell transduction.

Muller et al. [150] compared two different retroviral vector platforms, the lentiviral and alpharetroviral, both in combination with two different transduction enhancers (Retronectin and Vectofusin-1). They explored different NK cell isolation techniques (NK cell enrichment and CD3/CD19 depletion) to identify the most efficacious methods for genetic engineering of NK cells. Results from their study demonstrated that transduction of NK cells with RD114-TR pseudotyped retroviral vectors, in combination with Vectofusin-1 was the most efficient method to generate highly cytotoxic CD19-CAR-NK cells. In addition, retronectin was potent in enhancing lentiviral/VSV-G gene delivery to NK cells, but not alpharetroviral/RD114-TR.

Very recently, nanobodies have been used to generate CD38 CAR-NK cells. These molecules consist of the single variable heavy chains (VH) of Abs that are naturally present in llamas and dromedaries [151]. CD38-specific nanobodies have been derived from immunized llamas and fused by gene synthesis to the signal sequence of VH, hinge region, and trans-membrane domain of human IgG4, linked to the Immunoreceptor Tyrosine-based Activation Motif (ITAM) sequences of human CD28, 4-1BB and CD3ζ C-terminal domain. NK-92 cells were retrovirally transduced to stably express this construct [138]. Although these studies showed promising results, additional investigations should be performed to provide a basis for the clinical development of novel therapeutics.

Amphotropic retroviral vectors have also been used to generate CAR-NK cells. The protocols followed by Esser [139] and Romanski [128] described the generation of a construct with the scFv of ch14.18 anti-GD2 chimeric Ab or of anti-CD19 Ab, respectively. In both studies, scFv was combined with IgVH signal peptide, myc tag, the hinge region of CD8α and trans-membrane/intracellular domains of CD3ζ chain. These constructs were inserted in a pLXSN retroviral vector transduced in FLYA-JET packaging cells by electroporation. The transduced cells released the amphotropic retroviral vectors in the supernatant which was subsequently used to transducer NK-92 cells [128,139]. This kind of approach revealed that CAR expression by gene-modified NK cells facilitated effective recognition and elimination of established GD2- or CD19-expressing cells, which were resistant to parental NK-92 cells. In the case of intrinsically NK-sensitive tumor cell lines, the authors observed an evident increase in killing activity of retargeted NK-92 cells, strictly dependent on specific recognition of the target antigen. Kailayangiri et al. tried to enhance the potency of this therapeutic approach by incorporating several levels of optimization such as CAR design with integration of co-stimulatory molecules and extension of the cytokine support during in vitro expansion of NK cells. They combined scFv of the anti-GD2 Ab, 14.G2a, with the trans-membrane domain of CD28 and signaling domains of 4-1BB and CD3ζ in the SFG retroviral vector. Phoenix-ampho cells were transiently transduced with this vector to produce viral particles necessary to infect the packaging cell line FLYRD18, which is a stable retroviral producing cell. The resulting supernatants were used to transduce NK cells [152]. Although authors reported that CAR-redirected NK cells were unable to prevent or reduce tumor growth in pre-clinical models, they identified non-classical MHC class I molecule, HLA-G, as key factor in the tumor immune escape from NK cells. Similarly, Liu et al. produced CD19 CAR-NK using SFG retroviral vector carrying anti-CD19 scFv combined to CD28 trans-membrane domain, CD28/CD3ζ signaling domains, interleukin-15, and inducible caspase 9 as a safety switch. These CAR-NK cells were administered in patients with relapsed or refractory CD19^+^ cancers in phase-I/II clinical trials and exciting results were reported. In particular, in 73% of patients, the infused CAR-NK cells expanded and persisted at low levels for at least 12 months. Interestingly, administration of CAR-NK cells was not associated with the development of CRS, neurotoxicity, or GvHD, and there was no increase in the levels of inflammatory cytokines, including interleukin-6, over baseline. The maximum tolerated dose of CAR-NK cells was not reached [153]. Different methods of engineering NK cells with CAR construct and the potential sources of scFv and NK cells are shown in Figure 2.

## 5. Pre-Clinical and Clinical Applications of CAR Engineered NK Cells

### 5.1. Solid Tumors: Pre-Clinical Studies

#### 5.1.1. Breast Cancers

TNBC, representing ~15% of globally diagnosed breast cancers, is typically an incurable malignancy due to the lack of targetable surface antigens for development of effective therapy. To address the unmet need for TNBC treatment, Hu et al. [145] recently defined TF as a useful surface target expressed in 50–85% of TNBCs and developed a 2nd generation TF-targeting Ab-like immunoconjugate (called L-ICON) for the treatment of TNBC. Moreover, they developed the TF-CAR-NK cells which co-expressed CD16 and the Fc receptor (FcγIII) to mediate ADCC and tested them against TNBC as single agent therapy and in combination with L-ICON. Pre-clinical results demonstrated that TF-CAR-NK cells alone killed TNBC cells, but the cytotoxicity was enhanced with L-ICON. Such effects were replicated against tumors established by the injection of patient-derived cells into the xenograft mouse models.

Another approach to fight breast cancer is using redirected expanding NK-92 cells against the tumor-associated ErbB2 (HER2), a tissue antigen over-expressed in several human tumors, including breast, ovarian, stomach and prostate cancer [154]. Following GMP-compliant procedures, Moasser et al. generated a stable clonal cell line expressing a humanized CAR based on ErbB2-specific αβ, FRP5, harboring CD28 and CD3ζ signaling domains (CAR 5.28.ζ). These NK-92/5.28.ζ cells efficiently lysed ErbB2-expressing tumor cells in vitro and exhibited serial target cell killing. More importantly, specific recognition of tumor cells and anti-tumor activity were retained in vivo, resulting in selective enrichment of NK-92/5.28.ζ cells in orthotopic breast carcinoma xenografts, and reduction of pulmonary metastasis in a renal cell carcinoma model.

Chen et al. investigated the anti-tumor activity of EGFR CAR NK cells in response to EGFR^+^ BC cells in vitro. Such response was greater than that of parental NK-92 cells, in terms of cytotoxicity and IFN-γ secretion, whereas similar responses were detected using EGFR^−^ cell lines as targets. Additional experiments were performed in vitro testing EGFR CAR NK cells in combination with oncolytic herpes simples virus 1 (oHSV-1), demonstrating a synergistic effect in the induction of EGFR^+^ cell lysis [127]. Pre-clinical models were generated by inoculating MDA-MB-231 BC cells into the brain of non-obese diabetic SCID gamma (NSG) mice to study the different therapeutic strategies for patients with brain metastasis originated from breast tumors. After 10 days (of single therapy) or 15 days (of combination therapy) mice were injected intra-tumorally with EGFR CAR NK cells and/or oHSV-1. A synergistic effect was suggested since combined therapy generated better results in terms of tumor size reduction and prolonged survival of the treated mice [127].

#### 5.1.2. Glioblastoma (GB)

GB is the most common and aggressive primary brain tumor in adults which is currently incurable. Although GB tumors are frequently infiltrated by NK cells, such cells are actively suppressed by the GB cells and the GB tumor microenvironment. Interestingly, ex vivo activation with cytokines can restore cytolytic activity of NK cells against GB, indicating the NK cells potential for GB adaptive immunotherapy. Correspondingly, CAR-NK cells have been generated for GB immunotherapy [155].

Zhang et al. showed that HER2 was moderately to highly expressed in the majority of human GB and demonstrated that ErbB2 CAR-NK cells obtained from NK92 cells selectively recognized and lysed LN-319, LNT-229 and LN-428 ErbB2^+^ GB cell lines in vitro as well as ErbB2^+^ primary GB cells derived from patients [129]. Pre-clinical studies using immunocompetent mice as well as highly immunodeficient animals subcutaneously or orthotopically injected with GB cells, revealed that ErbB2 CAR-NK-92 cells were able to hinder the tumor growth and prolong the mice survival. Moreover, GB-specific Abs were detected in ErbB2 CAR-NK cells-treated mice suggesting that the released cytokine may trigger a specific anti-tumor immunity [129].

#### 5.1.3. Colorectal Cancers

Colorectal cancer accounts for approximately 10% of all annually diagnosed cancers and cancer-related deaths worldwide. It is the second most common cancer diagnosed in women and third most in men. With continuing progress in developing countries, the incidence of colorectal cancer worldwide is predicted to greatly increase [156]. Many TAA and tumor-specific antigens have been used as target structures for the development of colorectal cancer therapies or vaccines such as melanoma associated antigen (MAGE), vascular endothelial growth factor receptor 1 and 2 (VEGFR-1 and VEGFR-2), EpCAM and EGFR.

EpCAM is a type I trans-membrane glycoprotein, which was originally identified as a TAA due to its high expression in rapidly growing epithelial tumors. Zhan et al. constructed EpCAM-specific second^−^generation CAR that was transduced into NK-92 cells by lentiviral vectors to be used alone or in combination with the multi-kinase inhibitor, regorafenib. In a mouse model with human colorectal cancer xenografts, Zhang et al. demonstrated that the CAR-NK-92 cells specifically recognized EpCAM^+^ colorectal cancer cells, released cytokines such as IFN-γ, perforin and granzyme B, and exerted cytotoxic activities. In addition, the growth suppression of combination therapy with regorafenib and CAR-NK-92 cells was greater than that of monotherapy with CAR-NK-92 cells or regorafenib. Thus, such combinational therapeutic approach was introduced as a novel strategy to treat colorectal cancer and to enhance the therapeutic effects of CAR-modified immune effector cells for solid tumors [141].

Han et al. tested EGFR CAR NK cells against EGFR^+^GB cell lines. A significantly higher response was observed using EGFR CAR NK cells compared to parental NK-92 cells in terms of cytotoxicity and IFN-γ production against EGFR^+^ Gli36dEGFR, U251, and LN229 cells [132]. Notably, similar data were obtained using patient-derived GB cells as target, confirming the recognition of primary GB cells by EGFR CAR NK cells. Pre-clinical studies were carried out using orthotopic glioma model established by inoculating NSG mice intracranially with wtEGFR-expressing U251cells or EGFRvIII-expressing GB30 glioma cells. Mice subsequently treated with intra-tumoral injection of EGFR CAR NK cells displayed a significantly reduced tumor growth and prolonged survival, compared to mice inoculated with parental NK-92 cells, thus confirming a specific targeting of tumor cells by EGFR-specific NK CAR [132].

#### 5.1.4. Ewing Sarcoma

Ewing sarcoma is the second most frequent bone tumor of childhood and adolescence that can also arise in soft tissue. It is a highly aggressive cancer with a survival of 70–80% for patients with standard-risk and localized disease and ~30% for those with metastatic disease [157].

Kailayangiri et al. have tested NK cell responses in vitro to different Ewing sarcoma cell lines, including VH-64, CADO-ES-1, DC-ES-6 and MS-PES-4 [152]. They observed that even untransduced and activated NK cells respond to these cell lines by expressing CD25 and releasing IFN-γ and TNF-α, resulting in a significant reduction in the viability of tumor cells. However, this response was greatly enhanced using GD2 CAR-NK cells, suggesting an efficient targeting of the GD2 antigen by GD2 CAR-NK cells. Several in vivo experiments were performed using NSG mice that were intraperitoneally or subcutaneously inoculated with VH-64 GB cells and subsequently treated with GD2 CAR-NK cells. As mentioned before, these experiments revealed no significant inhibition of the tumor growth. A possible explanation may be related to a significant up-regulation of the immunosuppressive molecule, HLA-G, in tumors, which was triggered by NK cells administration. Accordingly, an up-regulation of CD85j expression, a ligand of HLA-G, was observed on GD2 CAR-NK cells and parental activated NK cells upon co-culture with Ewing sarcoma cell lines. Interestingly, HLA-G^−^ tumor cell lines were efficiently lysed by activated NK cells in vitro and perhaps became resistant to NK cell-mediated lysis when transduced with HLA-G1. This findings highlighted the role of HLA-G in abrogating the in vivo CAR-NK cell anti-tumor activities and provides a rationale for therapeutic strategies using anti-HLA-G neutralizing Abs in combination with GD2 CAR-NK cells for patients with Ewing sarcoma [152].

#### 5.1.5. Hepatocellular Carcinomas

Hepatocellular carcinoma (HCC) is an aggressive cancer with limited treatment options. Although CAR-NK cells represent a promising immunotherapeutic modality for this cancer, their potential application have been barely explored in HCC. GPC3 is highly expressed in HCC and other tumor types, while its level is undetectable in normal tissues which rationalizes its use as an immunotherapeutic target. Recently Min et al. developed GPC3-specific NK cells and explored their potential in the treatment of HCC [142]. The NK-92/9.28.ζ cell line was established by engineering NK-92 cells with 2nd generation GPC3-specific CAR. NK-92/9.28.ζ cells showed a potent anti-tumor activity against multiple HCC xenografts with both high and low GPC3 expression, but not in those without GPC3 expression. An evident infiltration of NK-92/9.28.ζ cells, a decreased tumor proliferation and an increased tumor apoptosis were observed in the GPC3^+^ HCC xenografts. Since efficient retargeting of primary NK cells was achieved, the authors provided the background to justify clinical translation of this GPC3-specific NK cell-based therapeutic as a novel treatment option for patients with GPC3^+^ HCC [142].

### 5.2. Engineered NK Cells for Hematological Malignancies

#### 5.2.1. Leukemias and Lymphomas

Leukemias and lymphomas represent a broad range of hematological malignancies, characterized by clonal growth and dysfunction of lymphoid and myeloid cells at different stages of maturation and commitment [158,159].

Functional studies on CAR-NK cells targeting B cell malignancies have been initially carried out in vitro. Boissel et al. demonstrated that CD20 CAR-NK cells recognize and kill leukemic blasts from chronic lymphocytic leukemia (CLL) patients in vitro. This killing was more efficient than ADCC induced by anti-CD20 mAbs rituximab or ofatumumab [140]. Studies on CD19 CAR-NK cells revealed an increased response to CD19^+^ targets, compared to parental NK cells, in terms of cytotoxicity, degranulation and release of IFN-γ and TNF-α. In addition, CD19 CAR-NK cells secreted IL-15 (encoded by viral vector) in response to CD19^+^ cell targets. CD19 CAR-NK cells were also generated from CLL patients’ PB cells and tested against CLL cells. However, the cytotoxicity of these cells was very low, probably due to the expression of ligands for inhibitory molecules, such as HLA-E [153]. Boissel et al. [140] tested CD19 CAR-NK cells in a xenograft model of B-ALL using NSG mice and demonstrated that: i) tumor growth was almost abrogated, and ii) malignant cells were nearly undetectable in PB, femur, or spleen. Since B-ALL cells predominantly localize in the BM mimicking the minimal residual disease, the authors treated mice with a single intra-femoral injection of CD19 CAR-NK cells at 100 days post-tumor challenge and observed a potent anti-tumor response, with undetectable level of leukemia cells in the femur.

Liu et al. introduced IL-15 in the CD19 CAR construct expressed by NK cells and reported that IL-15 co-expression not only limited the tumor growth and prolonged the mice survival, but also increased in vivo persistence and recruitment to the tumor sites of CD19 CAR cells [153].

CD19 CAR-NK cells were recently tested in a phase-I/II clinical trial (NCT03056339) on 11 patients affected by non-Hodgkin’s lymphoma or CLL. Patients were subjected to lympho-depleting chemotherapy with fludarabine and cyclophosphamide and then infused with CD19 CAR-NK cells at increasing doses. Nine patients received CAR-NK cells partially matched with recipient HLA genotype, whereas two patients were treated with CAR-NK cells with no information of HLA-matching. When possible, NK cells were selected based on the presence of KIR mismatch with the recipient, to enhance the intrinsic NK cell activity against tumor cells. The infusion of CAR-NK cells was safe and well tolerated, with no CRS, neurotoxicity, hemophagocytic lymphohistiocytosis, or GvHD. At median follow-up (13.8 months) the authors observed an objective response in 8 patients (7 with complete response and one with complete remission) occurred in the first month post-treatment. CD19 CAR-NK cells expanded in vivo from day 3 to14 and persisted after 12 months. Notably, responsive patients displayed a higher expansion of CAR-NK cells. Moreover, CAR-NK cells were present in the PB as well as in the BM and lymph nodes, suggesting their ability to home in secondary lymphoid organs. No alteration in the serum level of inflammatory cytokine was detected in treated patients, including IL-15 encoded by the construct, thus suggesting that CAR-NK cells did not released toxic systemic amounts of IL-15 post-infusion [160].

Pinzet al. characterized CD4 CAR-NK cells targeting T cell malignancies in vitro and in pre-clinical studies. They showed that CD4 CAR-NK cells killed CD4^+^ T cells from cord blood in vitro except for hematopoietic precursors, confirming sole recognition of mature CD4^+^ T cells. Pre-clinical studies revealed that CD4 CAR-NK cells reduced tumor burden and prolonged overall survival, thus demonstrating therapeutic effects in vivo [144]. Nonetheless, there is no available data from clinical trials using this type of CAR-NK cells.

CD33 CAR-NK cells were produced in a GMP facility to be tested in a phase-I study for the treatment of patients with acute myeloid leukemia (AML) [146]. A 3rd generation CAR lentiviral construct containing both CD28 and 4-1BB co-stimulatory molecules, with an Fc fragment inserted between the CD33 scFv and CD28 was used to transduce NK-92 cells. The transduction efficiency was over 90% and the transduced cells were successfully expanded in a GMP facility. The CD33 CAR-NK cells were administered to three AML patients undergoing salvage chemotherapy (NCT02944162). Limited side effects (fever and grade I CRS) were detected in one patient with a temporary increase in plasma level of IL-6, IL-10, and IL-17A. One month post-CD33 CAR-NK infusion, one patient displayed absence of leukemic blasts in the BM, with minimal residual disease of 1.7 × 10^−3^ cells with 88.2% CD33^+^ expression. Unfortunately, this patient showed relapse four months after the last CD33 CAR-NK cell infusion, and the other two patients did not respond to the treatment. Nevertheless, this phase-I/II study demonstrated that infusion of CD33 CAR-NK cell is safe and well tolerated [146].

Finally, a few additional phase-I/II clinical studies that are in progress are listed below:“CAR-modified pNK cells for CD7^+^ relapsed or refractory leukemia and lymphoma” (NCT02742727) using NK-92 cells engineered with anti-CD7 attached to TCRζ, CD28, and 4-1BB signaling domains.“PCAR-119 bridge immunotherapy prior to stem cell transplant in treating patients with CD19^+^relapsed or refractory leukemia and lymphoma” (NCT02892695) based on the use of NK-92 cell line engineered to contain anti-CD19 attached to TCRζ, CD28, and 4-1BB signaling domains.

#### 5.2.2. Multiple Myeloma

MM accounts for ~10% of all hematologic malignancies [161]. It is characterized by the clonal proliferation of plasma cells that produce M-proteins leading to various impaired immune functions [162].

CD38 nanobody CAR-NK cells have been first tested in vitro for their ability to recognize and kill CA-46 Burkitt lymphoma cells. Hambachet al. demonstrated that NK-92 cells transduced with anti-CD38 nanobody CAR, efficiently lysed CA-46 cells [138]. Subsequently, they exposed BM samples from MM patients containing CD38^hi^CD56^hi^CD319^hi^ myeloma cells to the CD38-specific nanobody CAR-NK cells and showed significant lysis of primary myeloma cells compared to untransduced NK cells [138].

CS1 (also known as CD319, CRACC and SLAMF7) is expressed on leukocytes and NK cells, but not in any major body organs. CS1 is highly expressed in MM and may be a potential target for treating MM [163]. To take advantage of CS1 selective expression, CS1 CAR-NK cells were generated and tested against MM cells. CS1 CAR-NK cells were able to respond to CS1^+^ IM9 and L363 MM cells more efficiently than parental NK-92 cells, in terms of cytotoxicity and IFN-γ secretion. Similar results were obtained using patient-derived CD138^+^ MM cells as target. Pre-clinical studies in xenograft orthotopic MM model demonstrated that CS1 CAR-NK cells significantly reduced tumor volume and prolonged survival. This study paved the way for future clinical studies using CS1 CAR-NK cells as immunotherapy for MM patients [143].

To date, one phase-I/II clinical study (NCT03940833) which aims at investigating the safety and feasibility of B cell maturation antigen (BCMA) CAR-NK 92 cells in patients with relapsed and refractory MM is ongoing. These engineered NK cells were designed based on the premises that BCMA is over-expressed on the surface of malignant plasma cells and anti-BCMA CAR enabled the NK-92 cells to recognize and kill MM cells in pre-clinical models.

#### 5.2.3. Solid Tumors: Clinical Trials Using Engineered NK Cells

To date, only a few phase-I/II clinical trials are in recruiting status which are listed as following.

“Clinical research of ROBO1 specific CAR-NK cells on patients with solid tumors” (NCT03940820). The purpose of this study is to evaluate the safety and effectiveness of cell therapy using ROBO1 CAR-NK cells in treating different types of solid tumors. The acceptance criteria for patients are the ROBO1 expression in malignancy tissues detected by immuno-histochemistry and the diagnosis of advanced solid tumor.“Study evaluating the efficacy and safety of CAR-modified PB NK cells in MUC1^+^ advanced refractory or relapsed solid tumors” (NCT02839954).“Clinical research of ROBO1-specific BiCAR-NK cells on patients with pancreatic cancer” (NCT03941457).

## 6. Conclusions

So far, numerous pre-clinical studies have confirmed the anti-tumor activities of γδ T cells in targeting various malignancies. Clinical studies using γδ T cells have demonstrated safety and proposed a potential for developing allogenic ‘off-the-shelf’ therapies. A major hinderance in the clinical application of γδ T cells is their poor ex vivo expansion. To address this issue, strategies of genetic engineering may enhance the proliferation of multiple γδ T cells subsets and improving the in vivo survival and persistence. Transduction approaches expressing cytokine or growth factor receptors may improve the required characteristics and function. For example, expressing IL-7 receptor by γδ T cells may enhance their proliferation and survival. Modern high-throughput techniques have provided a vast opportunity to better understand the different γδ T cells subsets at single-cell level. Advancement in the expansion of γδ T cells and enhancing their intrinsic anti-tumor activity using genetic manipulations, will provide novel adaptive cell therapy with the potential of allogenic use.

Recent studies using CAR-NK cells targeting tumor antigens have demonstrated that these cells can be efficiently obtained from normal donors or commercially available NK cell lines. The in vitro recognition of tumor cell lines carrying the specific antigen by CAR-NK cells has been confirmed by numerous studies. Pre-clinical studies have demonstrated that CAR-NK cells are effective in cancer therapy, by limiting tumor growth and promoting survival of mice. Such therapeutic effects were observed in both hematological and solid malignancies models. So far, few clinical trials have evaluated the therapeutic efficacy of CAR-NK cells in patients with hematological malignancies (i.e., AML and CLL). However, only in CLL patients, CAR-NK cells (specific for CD19) triggered a partial or complete response. Nevertheless, phase-I/II studies to date, have demonstrated that administration of CAR-NK cells is safe and well tolerated. Lastly, 26 clinical trials using CAR-NK cells are still ongoing for patients with B cell lymphoma, leukemias, MM and different solid tumors (including hepatocellular carcinoma, pancreatic cancer, and glioblastoma). Since most of these trials are in the early stages (recruiting or not yet recruiting) clinical data are not available yet (www.clinicaltrials.gov).

In conclusion, engaging the non-conventional niches of immune system such as γδ T cells and NK cells represents an ideal strategy to reduce the systemic stress and, therefore, the risk of CRS that have been known to be a life-threatening complication after CAR αβ T cell therapies. So far, clinical experiences with engineered γδ T and NK cells have proven the safety and lack of on-target off-tumor toxicity which provides the opportunity to target those antigens that, due to toxicity concerns, have been off-limits to date. However, a deeper investigation of the therapeutic role of engineered γδ T cells and NK cells for cancer therapy is required.

## Figures and Tables

**Figure 1 cells-09-01757-f001:**
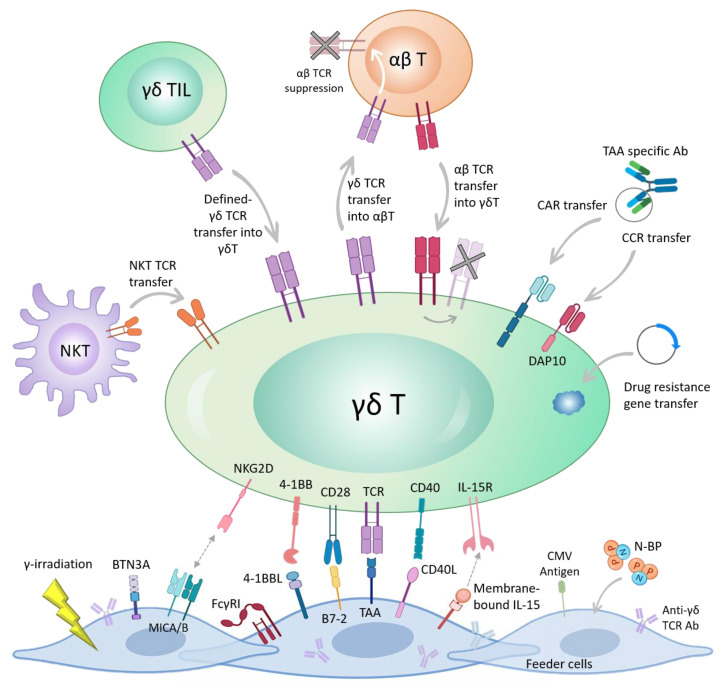
Different genetic engineering strategies for harnessing the anti-tumor activity of γδ T cells. TCR-based engineering methods include TCR gene transfer from αβ to γδ T cells and vice versa. Once new TCR is introduced into αβ T cells, the expression of endogenous αβ chains will be suppressed. Synthetic CAR and chimeric co-stimulatory receptor (CCR) DNA can be transferred to γδ T cells to redirect γδ T cells toward a specific tumor antigen. CCR is similar to CAR, but lacks the CD3ζ domain, thus full activation of γδ T cells requires the support of additional co-stimulators (e.g., endogenous CD3). Transduction with drug resistance gene is another strategy aiming for rendering γδ T cells resistant to chemotherapy drugs. NKT receptor can be transferred into γδ T cells. The TCR from TILs γδ T cells may be transferred into new γδ T cells to endow anti-tumor response. Artificial APC or feeder cells may be treated with nitrogenous bisphosphonates (N-BP) to better activate γδ T cells. In addition, aAPCs may be engineered to express anti-γδ TCR Abs or different co-stimulatory molecules/ligands such as 4-1BB, B7-2, CD40L, membrane-bound IL-15 and cytomegalovirus (CMV) peptide antigens. Normally γ-irradiation is performed to hinder the aAPC growth. Stressed aAPCs, due to γ-irradiation or exposure to N-BP, up-regulate MICA/B and Butyrophilin (BTN) proteins which bind to NKG2D and activate γδ T cells. A tumor-associated antigen (TAA) may be also expressed by aAPC to selectively expand the TAA-reactive γδ T cells.

**Figure 2 cells-09-01757-f002:**
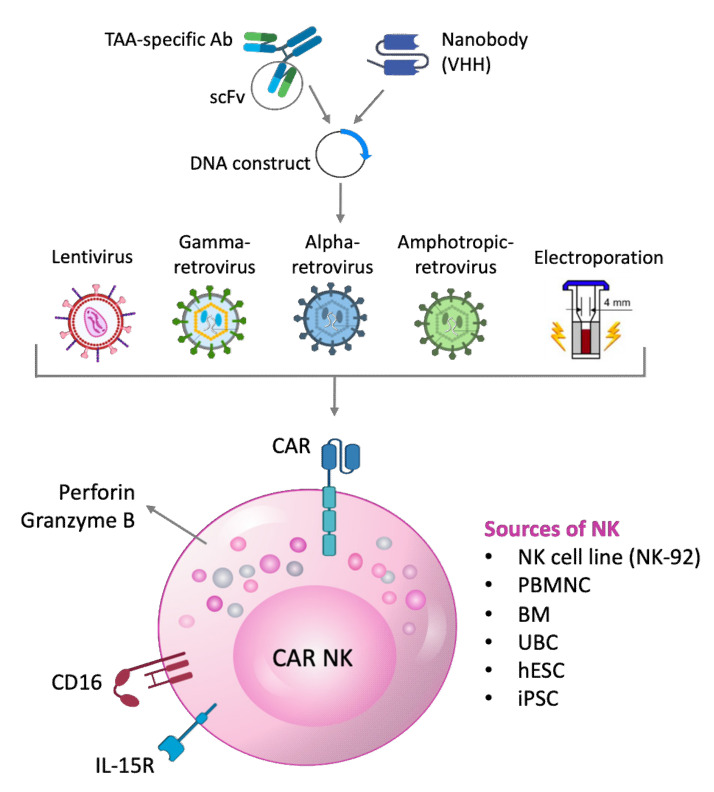
Methods of generating CAR-NK cells. NK cells can be obtained from various sources of an autologous or allogenic donor. The CAR construct, consisting of the scFv sequence (derived from either a tumor antigen/TAA-specific Ab or nanobodies) linked to activatory domains, is made via molecular cloning or gene synthesis techniques. This construct is expressed in NK cells using viral or non-viral transduction (i.e., electroporation). When exposed to the tumor antigen-expressing tumor cells, CAR-NK cells become activated and release cytokines and cytolytic agents such as perforin and granzyme B. Along with CAR, other molecules may be co-transduced to improve persistence, migration, and/or cytotoxicity of CAR-NK cells. Among them, the IL-15 cytokine receptor and the FcγRIII CD16 (in NK-92MI lacking CD16) which triggers ADCC. PBMNC: peripheral blood mononuclear cells; BM: bone marrow; UBC: umbilical cord; hESC: human embryonic stem cells; iPSC: induced pluripotent stem cell.

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
