# Peer review of "Engineering the Bridge between Innate and Adaptive Immunity for Cancer Immunotherapy: Focus on γδ T and NK Cells"

_cells, 2020, doi:10.3390/cells9081757_

Round 1
Reviewer 1 Report
Overall, the topic is interesting. Minor changes suggested for this review article. I am attaching a work document that has my specific comments and suggestions

Reviewer 2 Report
The manuscript by Morandi et al. comprehensively reviews the current research on NK and gamma-delta (gd) cell engineering. The authors must be commended for the breadth and depth of their work. Overall this manuscript can become a reference for a large readership interested in cellular immunotherapy.
Minor modifications/additions would nonetheless improve the manuscript.
1- The use of the acronym TAA (tumor-associated antigen) is misleading. The cell-surface molecules targeted by the antibody moiety of CARs do not fulfill the commonly used definition of TAA. For example, CD19 is similarly expressed by the malignant and healthy B cells, leading to B-cell depletion in CAR T-cell therapy. It is not per se a tumor antigen. A careful revision of the manuscript is required to assess whether the use of TAA is appropriate.
2- Line 117. Please mention that the success of CAR is solely demonstrated for B-cell neoplasms, not hematological malignancies in general.
3- Is defining gd T-cell activation as MHC-independent accurate (line 142) ? Activation is certainly independent of classical MHCI and MHCII molecules, but non-classical MHC can be antigenic targets (among others) it appears. A short paragraph summarizing clearly what is known and unknown about antigen recognition by gd T cells would dissipate the confusion.
4- Line 237. The authors should briefly summarize the clinical data using unmodified gd T cells. Despite the marked enthusiasm and rationale to use these cells, substantial and durable responses remain limited (if not anecdotal). In fact, one may argue that unless engineering or combination therapy is envisaged, the full potential of gd T cells may not be harnessed for cancer immunotherapy.
5- Line 267. A short description of the mechanisms underlying the antigen cross-presentation capabilities of gd T cells would highlight this relevant but largely unknown function of this lymphoid subset.
6- The author briefly mention that gd T cells do not seem to exhaust after ex vivo expansion. What is known about gd T-cell exhaustion? How was this studied? Can we only rely on phenotypes and ex vivo functional assays? This issue is critical given the need for ex vivo propagation of these cells for therapy. From the information provided, one cannot conclude that the marked fold expansions observed does not lead to some kind of impairment after adoptive transfer (in vivo). If such data exist, they should be described, if not, it should be mentioned that more research is required to evaluate whether the concepts of T-cell dysfunction as described for classical T cells (exhaustion, terminal differentiation, activation-induced cell death, senescence) apply to gd T cells.
7- Line 377. gd T cells and NK cells are proposed as ideal “third party” effectors, which is well-justified. Third party cells are short-lived which may be a limitation (persistence of CAR T cells is a key variable for success, particularly in ALL). This should be described, as well as the possible approaches to circumvent this limitation.
8- The text should be reviewed for typos.
Reviewer 3 Report
Morandi and colleagues present an extensive review about the use of gd T and NK cells in cancer immunotherapy. The organization of the review is straightforward, they start by an introduction to the field of adaptive cellular therapy (TCR and CAR), they follow with gd T cells and finish with NK cells and their application in specific cancer treatments. The field is vast and this work will give a good overview of the topic to the readers.
The reviewer has some comments that can easily be fixed but should be taken in account:
- citations 13 and 14 do not seem to be the correct references: 13 should be replaced by 16 and 17 (and eventually PMID 20042572) which are the first demonstrations of mispairing and 14 should be replaced by PMID: 24114521. Citation 14 refers to the improved TCR immunotherapy by increasing the presence of CD3 (here a murine construct).
- The paragraph about endogenous TCR disruption (l. 64-91) is confusing, maybe it should be moved as a sub-chapter, for example some of the citations such as (26) depict CAR T cell work under the TCR sub-chapter. In addition, one of the main references for the subject is not found and should be added: PMID: 28225754.
- L.234 strange signs
- L.256: “sonce” should be “since”
- L. 285 strange sign
- L. 344 the two clinical trials cited refer to NK-based trials, this could be confusing since the chapter discusses gd T cells.
- Chapter 4 on NK cell engineering omits a subchapter about TCR expressing NK: PMID: 30665853 and PMID: 31054264 (and eventually PMID: 28878363). These works should be discussed in this review
- L.595 “Boisell” should be “Boissel”
- L. 620 “Authors” should be “authors”
